# Lack of Significant Effects of Glyphosate on Glyphosate-Resistant Maize in Different Field Locations

**Vitor Simionato Bidóia** [1], **José Cristimiano dos Santos Neto** [2], **Cleber Daniel de Goes Maciel** [2],
**Leandro Tropaldi** [3], **Caio Antonio Carbonari** [4], **Stephen Oscar Duke** [5,*] **and Leonardo Bianco de Carvalho** [1]

1   School of Agricultural and Veterinarian Sciences, São Paulo State University (UNESP),
    Jaboticabal, SP 14884-900, Brazil; vitorsbidoia@gmail.com (V.S.B.); leonardo.carvalho@unesp.br (L.B.d.C.)
2   State University of the Central West (UNICENTRO), Campus Cedeteg, Guarapuava, PR 85040-167, Brazil;
    neto.buri@hotmail.com (J.C.d.S.N.); cmaciel@unicentro.br (C.D.d.G.M.)
3   College of Agricultural and Technology Sciences, São Paulo State University (UNESP),
    Dracena, SP 17900-000, Brazil; leandro.tropaldi@unesp.br
4   School of Agriculture, São Paulo State University (UNESP), Botucatu, SP 18610-034, Brazil;
    caio.carbonari@unesp.br
5   National Center for Natural Products Research, School of Pharmacy, University of Mississippi,
    University, MS 38677, USA
*   Correspondence: sduke@olemiss.edu

**Abstract:** Glyphosate-resistant (GR) maize is dominant in countries where it is grown. Significant, adverse effects of glyphosate application to GR maize have been reported, but few data from robust studies exist to determine if such effects are common. In this study, the effects of recommended application rates (single and sequential applications) were used on GR maize grown at two locations for one season and for two seasons in a third location. No significant effects of glyphosate on mineral content (N, P, K, Ca, Mg, S, Cu, Fe, Mn, and Zn) in leaves or grain, plant height, stem diameter, ear parameters, or yield were found at any location or in any growing season. Likewise, harvested grain quality, as determined by percent starch, protein, and total lipids, was unaffected by glyphosate treatment at any location. Neither glyphosate nor aminomethylphosphonic acid, the primary degradation product of glyphosate, were found in grain from any treatment at any location, except for 20 ng g$^{-1}$ of glyphosate found in grain from one season at one location. These results support the view that recommended applications of glyphosate have no significant effects on growth, grain composition, mineral content, grain quality, nor yield of GR maize.

**Keywords:** aminomethylphosphonic acid; glyphosate; herbicide; maize; mineral nutrition; seed quality; transgenic crop; yield; *Zea mays*

## 1. Introduction

Glyphosate (*N*-phosphonomethyl glycine)-resistant (GR) crops have been rapidly and widely adopted in North and South America. Despite the great success of these crops, some have claimed that glyphosate application to GR crops has significant adverse effects on them. One of the claimed effects is the alteration of plant mineral nutrition, including reductions in levels of both micro and macro nutrients in both soybean and maize [1–9]. Because glyphosate is known to chelate divalent metal cations [10], such an effect might be expected. However, more complete studies, especially with GR soybean [11–14], and to a lesser extent with GR maize [15,16], have found no effects on mineral levels.

Glyphosate has been reported to inhibit photosynthesis in GR crops [4,6] and to reduce yields [6,17,18]. However, the overall yield of three GR crops (soybean, cotton, and maize) in the USA has generally increased since they were introduced, with the same rate of yield increase as before their introduction [19,20].

Another adverse effect that has been claimed is increases in plant diseases (sometimes linked to claims of effects on mineral nutrition) [21,22]. Such effects would be expected

to affect yield and grain quality. This is not a well-researched claim, but a comprehensive study with effects of glyphosate on sudden death syndrome in GR soybeans [14] and another on the effects of glyphosate on Goss's wilt in GR maize [23] found no effects on the respective crop diseases. Several studies show that glyphosate can kill some plant pathogenic fungi in the GR crops that were reviewed in [24,25].

There is little information on the grain quality of glyphosate-treated GR crops, although several papers were published regarding the finding that the gene for glyphosate resistance in GR crops has no effect on grain composition, e.g., [26]. However, these early studies did not analyze seeds from GR crops treated with glyphosate, because at that time there was more concern about the effect of the transgene on crop yield and quality than the effect of glyphosate.

Glyphosate is the most used herbicide worldwide, and it is most used in GR crops. Considering the importance of GR crops and the massive volume of published research on glyphosate and GR crops, it is surprising that more research had not been conducted to determine whether the claims of adverse effects of glyphosate have validity, especially in maize, for which there are relatively few data compared to the data available for soybean. Here, we report the effects of recommended doses of glyphosate applied to GR maize crops as a single application or as two applications at three locations (one season for two locations and two seasons for the third) in Brazil. No adverse effects were found on growth, contents of macronutrients and micronutrients, ear parameters, grain composition, or yield.

## 2. Materials and Methods

### 2.1. Study Sites and Growing Conditions

Four field studies were conducted during two summer seasons in Brazil (seasons 2020–2021 and 2021–2022) in three sites: (i) Dracena (season 2020–2021, Lixisol with 14%-clay, 7% silt, 79% sand, and 2.1% loam, latitude 21°27′24″ S, longitude 51°33′23″ WGr and altitude 373 m); (ii) Guarapuava (seasons 2020–2021 and 2021–2022, Ferralsol with 56%-clay, 33% silt, 26% sand, and 3.6% loam, latitude 25°22′59″ S, longitude 51°33′13″ WGr and altitude 986 m); and (iii) Jaboticabal (season 2021–2022, Ferralsol with 42% lay. 33% silt, 25% sand, and 2.8% loam, latitude 27°47′10″ S, longitude 50°18′09″ WGr and altitude 890 m) counties, in the Sao Paulo state, Brazil.

Soil liming and fertilization were performed according to commercial recommendations based on soil analysis. Additional plant protection treatments with fungicides and insecticides were performed according to recommendations.

Maize was mechanically sown to a density of 70,000 plants ha$^{-1}$, using rows 50 cm apart. Experimental plots consisted of four planting rows all 12 m in length. Plant samples were collected from within the ten m$^2$ areas within the two central rows, and an inner 10 m length was used.

### 2.2. Materials

The seeds of a transgenic maize hybrid AG 8088 PRO2 (Agroceres, Santa Cruz das Palmeiras, SP, Brazil) were used. The transgenic events in this hybrid are MON80034 and NK603, which provide the Cry1A.105 and Cry2Ab2 and CP4-EPSPS genes, respectively, for insect and glyphosate resistance. A commercial glyphosate ammonium salt (783 g L$^{-1}$) formulation (Monsanto, São José dos Campos, SP, Brazil) equivalent to 720 g L$^{-1}$ of the free acid of glyphosate was used for all herbicide treatments. All other reagents used were analytical grade.

### 2.3. Experimental Design, Herbicide Application, and Weed Removal

For all experiments, we used a randomized block design with ten replicates and the following treatments: (1) herbicide-free control; (2) a single application of glyphosate at 980 g of acid equivalent per hectare (g a.e. ha$^{-1}$) at stage V5; and (3) a sequential application of glyphosate at 580 and 980 g of a.e. ha$^{-1}$ at stages V3 and V7, respectively. Control plots were protected from herbicide spray drift by sheets of plastic. Control samples were

collected from the two central rows of control plots to minimize the chances of sampling from plants that might have been exposed to spray drift. Herbicide treatments were applied with backpack sprayers pressurized with $CO_2$ and equipped with four flat-fan model 80.02E VS nozzles (TeeJet, Cotia, SP, Brazil), delivering a spray volume of 200 L ha$^{-1}$ at 200 kPa. All applications were performed in the early morning under no significant wind to avoid spray drift or under no cloudy conditions to avoid the washing off of spray with rain; the environmental conditions were moderate temperatures and relative humidities.

Weeds were removed shortly after emerging by hand weeding to prevent weed interference.

### 2.4. Growth Variables

Measurements of plant stem diameters were performed at the maize pre-flowering time using a graduated (1-mm) meter stick and a digital caliper (0.1-mm), respectively. Plant height was determined from the ground surface level to the insertional position for the newest leaf. Stem diameter was measured at 3 cm above the ground surface at the smallest and greatest dimensions, and the average stem diameter was calculated as the mean value of these dimensions. Ten samples from each treatment were used.

### 2.5. Foliar Mineral Content

The content of macronutrients (N, P, Ca, Mg, and S in g kg$^{-1}$) and micronutrients (Cu, Fe, Mn, and Zn in mg kg$^{-1}$) was determined via the same methods as described previously [16]. Five plants per replicate were used for mineral extraction. The leaf opposite and just below the ear was collected at the tassel stage (VT). Leaves were washed after collection. This was performed with sequential immersions into a neutral detergent solution, distilled water, and finally distilled water. After washing, leaves were air dried in a convection oven at 65 °C for 7 days. Dried materials were powdered with a Wiley mill with a 20-mesh steel screen and stored in glass bottles with silicon lids. Ground samples were extracted differently, depending on the nutrient. N and P contents were determined using a semi-micro Kjeldahl method and phosphor–vanadate–molybdate colorimetry, respectively. K, Ca, Mg, Cu, Fe, Mn, and Zn contents were determined via atomic absorption spectroscopy. The S content was obtained with a turbidimetric method. Ten samples from each treatment were used.

### 2.6. Crop Yield Variables

Crop yield variables were determined in ten plants per replicate, except for the estimated yield (ton h$^{-1}$). The ear insertion height was determined in centimeters using a graduated (mm) meter stick just before harvest. After ears were manually harvested, the number of rows of grain per ear and number of grains per row in the ear were counted. In sequence, the weight of one thousand grains and the total mass of grains per plant were determined with an analytical balance.

### 2.7. Grain Quality Variables

The content of starch, crude protein, and total lipids as a percent of the dry weight of the grain was determined as before [16]. For starch determination, grains samples of 5 g were used. Reduced sugars were determined via titration with mixed Fehling's solution, using methylene blue as an indicator. The content of reduced sugar was multiplied by 0.9 to calculate the starch content. For crude protein, nitrogen was extracted from 0.5 g samples using the Kjeldahl method, using sulfuric acid for digestion. The N content was multiplied by 6.5 (1% N corresponds to a 6.2 g protein to crude protein content). For total lipids, 5 g samples of grain were extracted using the Soxhlet method, using petroleum ether as the solvent. Total lipids were determined gravimetrically. Ten samples from each treatment were used.

### 2.8. Glyphosate and Aminomethylphosphonic Acid (AMPA) Residues in Grain

The content of glyphosate and aminomethylphosphonic acid (AMPA), the most common degradation product of glyphosate [25], in dried (65 °C for 7 days) grain was determined using a modification of a previously published method [27] and HPLC (Shimadzu, Kyoto, Japan) coupled to a triple quadrupole mass spectrometer (LS-MS/MS AB Sciex API 4500, Applied Biosystems, Foster City, CA, USA) with an electrospray ionization (ESI) source in the negative ionization mode to detect and quantify underivatized glyphosate and AMPA. One hundred mg of dried and powdered grain material was extracted for each sample. Ten samples from each treatment were used. Dried material was macerated, processing the extraction with the extraction solutions acetonitrile-water (50:50) [$v/v$] and methanol–water (50:50) [$v/v$]; it was then subjected to an ultrasonic bath and centrifugation. The supernatant was filtered and placed in an amber vial. Samples were analyzed via liquid chromatography and mass spectrometry (LC-MS/MS). The content of glyphosate and AMPA per gram of the dry sample was determined. Ten samples from each treatment were used.

### 2.9. Statistical Analysis

Estatistica® software (StatSoft, version 8.0, Tulsa, OK, USA) was used for statistical analysis to determine the 95% probability of effects. Data were tested for normality of residuals (Shapiro–Wilk and Kolmogorov–Smirnov tests) and homogeneity of variances (Levene and Bartlett tests). In sequence, the non-parametric statistical Kruskal–Wallis test was performed because many variables did not test for the normal distribution of residues and/or homogeneity of variances.

## 3. Results

### 3.1. Growth Variables

Growth variables just before flowering varied slightly with season and location, from 182 to 202 cm in plant height and from 18.8 to 21.8 mm in stem diameter, with no significant effects ($p > 0.05$) for the glyphosate treatments (Figure 1). There was no significant difference in either variable between locations or between seasons.

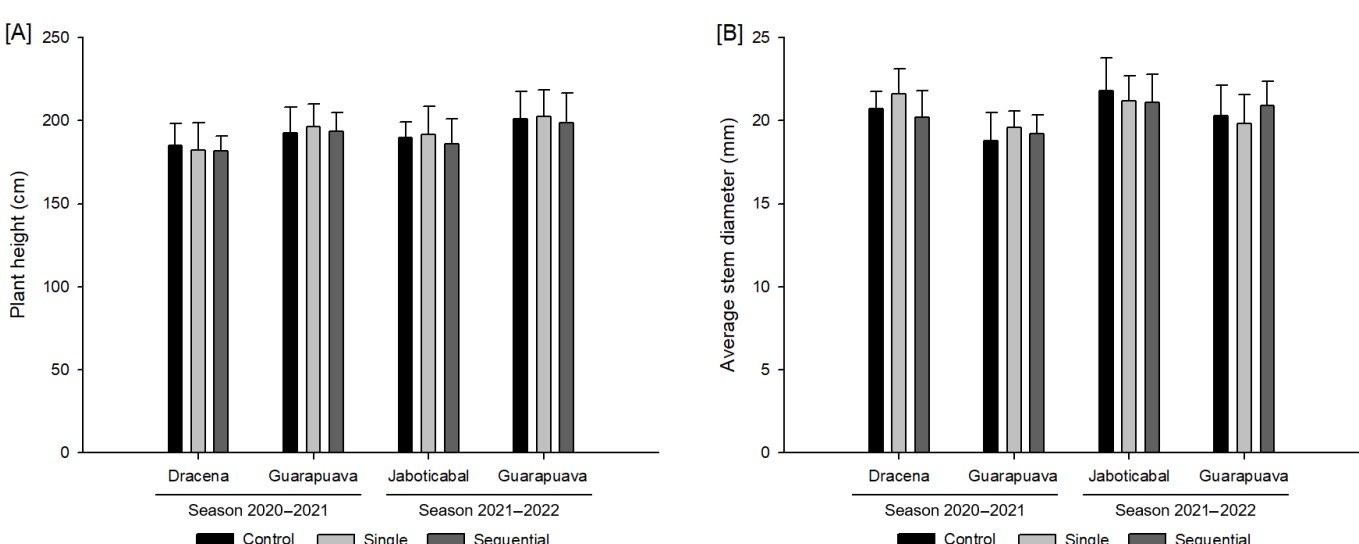

**Figure 1.** Plant height (**A**) and average stem diameter (**B**) at the pre-flowering time of GR maize grown with no glyphosate application (control), with a single application (980 g a.e ha$^{-1}$) at stage V5 or a sequential application of glyphosate at the V3 (520 g a.e ha$^{-1}$ and V7 (980 g a.e. h$^{-1}$) growth stages in three locations and two seasons. Error bars are one SE of the mean from ten replicates.

### 3.2. Foliar Mineral Content

Foliar elemental analysis varied among sites and seasons with no significant ($p > 0.05$) effect of glyphosate treatment (Figure 2). However, the content of macronutrients in foliar tissue from different sites varied from 12.6 to 16.3 g kg$^{-1}$ for N, 1.82 to 2.39 g kg$^{-1}$ for P, 20.9 to 29.5 g kg$^{-1}$ for K, 2.24 to 2.68 g kg$^{-1}$ for Ca, 2.89 to 3.21 g kg$^{-1}$ for Mg, and 2.54 to 2.98 g kg$^{-1}$ for S, and the micronutrient content between sites varied from 4.69 to 7.06 mg kg$^{-1}$ for Cu, 54.9 to 73.1 mg kg$^{-1}$ for Fe, 42.3 to 49.3 for Mn, and 9.69 to 11.05 for Zn. There was a significant difference only in leaf Cu content between locations (Guarapuava and Jaboticabal) in the season 2021–2022.

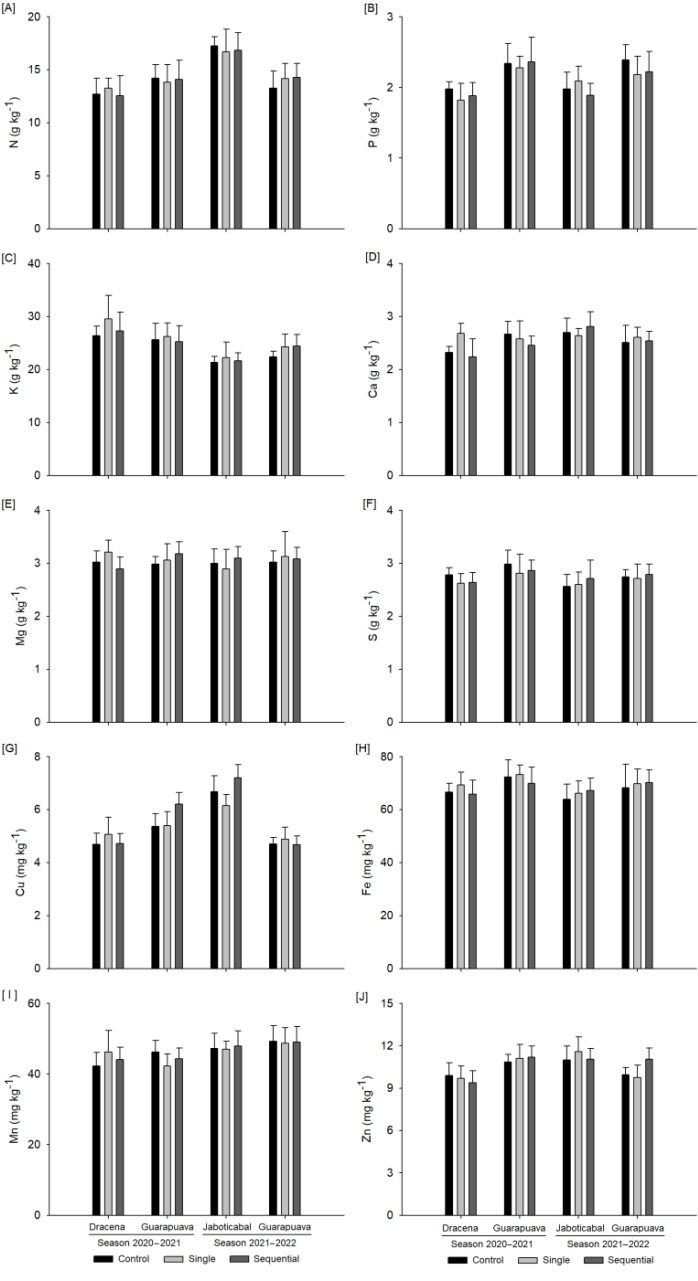

**Figure 2.** Content of macro-(N, P, K, Ca, Mg and S, respectively (**A–F**)) and micronutrients (Cu, Fe, Mn and Zn, respectively (**G–J**)) in leaf tissues of GR maize grown with no herbicide application (control), with a single application of glyphosate at 980 g of a.e. ha$^{-1}$ at stage V5 (single) and a sequential application of glyphosate at 520 + 980 g of a.e. ha$^{-1}$ at stage V3 and stage V7, respectively (sequential), in different locations and seasons. Error bars are one SE of the mean from ten replicates.

### 3.3. Crop Yield Variables

Crop yield variables varied among sites and seasons from 79.2 to 143.0 cm (ear insertion height), 13 to 19 units (number of rows of grain per ear), 31 to 39 units (number of grains per row), 263.5 to 356.5 g (weight of one thousand grains), 140.6 to 2012.0 g (total mass of grains), and 7.5 to 15.0 ton. ha$^{-1}$ (estimated grain yield), with no significant effect ($p > 0.05$) for the herbicide treatments tested in this research (Figure 3). There were significant differences in either variable between locations and/or between seasons for: ear insertion height, weight of one thousand grains, total mass of grains, number of grains pers row, and estimated grain yield (Dracena and Guarapuava in season 2020–2021, Dracena in season 2020–2021, and Guarapuava in season 2021–2022); for number of rows of grain per ear (Dracena season 2020–2021 and all others); and for estimated grain yield (Jaboticabal season 2021–2022 and Guarapuava in both seasons).

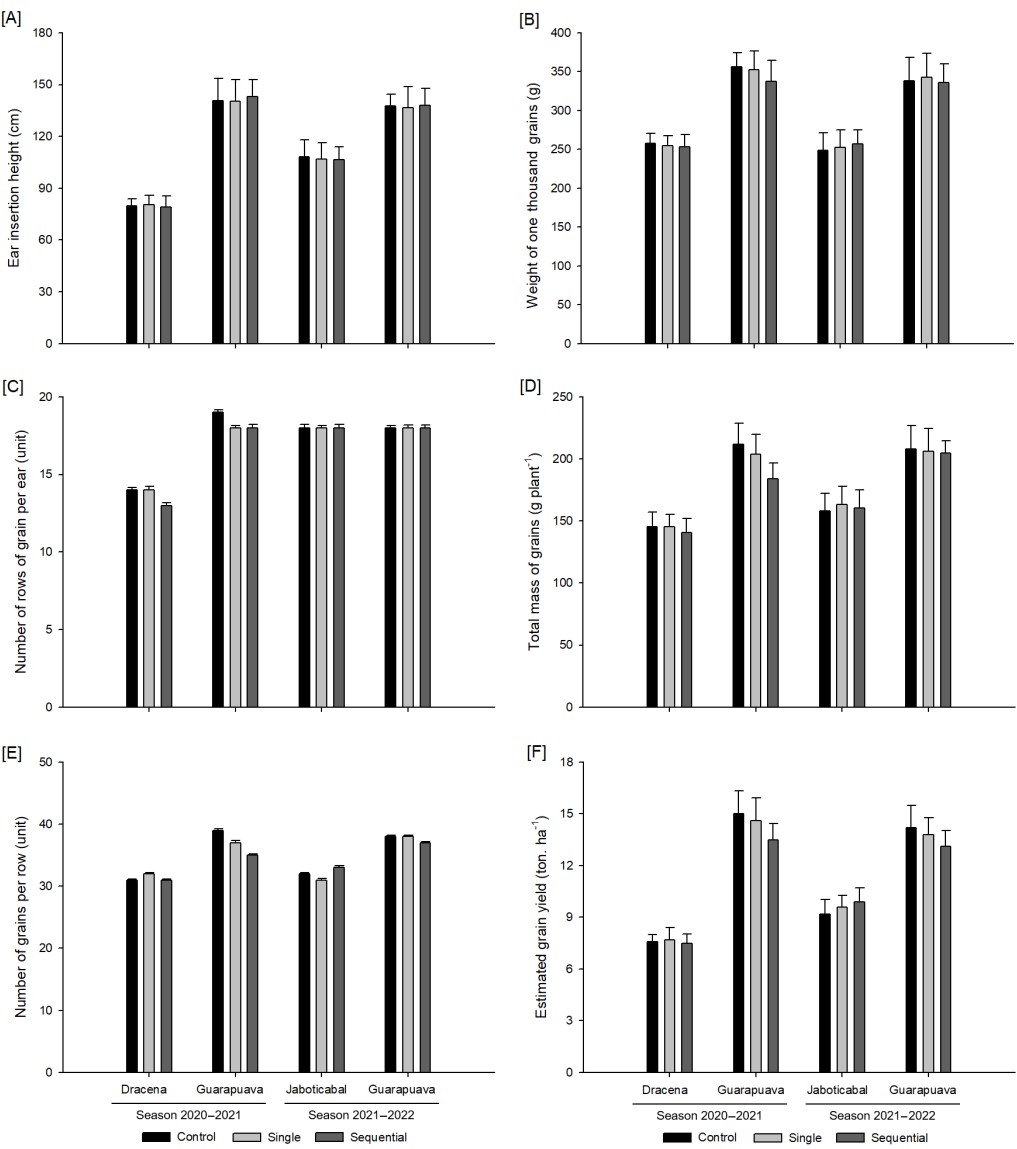

**Figure 3.** Ear insertion height (**A**), weight of one thousand grains (**B**), number of rows of grain per ear (**C**), the total mass of grains (**D**), number of grains per row (**E**), and the estimated grain yield (**F**) of GR maize grown with no herbicide application (control), with a single application of glyphosate at 980 g of a.e. ha$^{-1}$ at stage V5 (single) and a sequential application of glyphosate at 520 + 980 g of a.e. ha$^{-1}$ at stage V3 and stage V7, respectively (sequential), in different locations and seasons. Error bars are one SE of the mean from ten replicates.

### 3.4. Grain Quality Variables

Grain quality variables varied among sites and seasons (71.5 to 78% starch), 7.9 to 8.6% crude protein, 6.8 to 7.9% total lipids, 12.8 to 14.3 g kg$^{-1}$ for N, 2.02 to 2.22 g kg$^{-1}$ for P, 3.12 to 4.53 g kg$^{-1}$ for K, 34.2 to 42.9 g kg$^{-1}$ for Ca, 0.99 to 1.24 g kg$^{-1}$ for Mg, 0.90 to 0.98 g kg$^{-1}$ for S, 2.06 to 2.56 mg kg$^{-1}$ for Cu, 4.26 to 6.69 mg kg$^{-1}$ for Fe, 4.06 to 4.82 mg kg$^{-1}$ for Mn, and 13.6 to 15.9 mg kg$^{-1}$ for Zn, with no significant effect ($p > 0.05$) for the herbicide treatments tested in this research (Figures 4 and 5). There was a significant difference only in grain Fe content between locations and seasons (Dracen season 2020–2021 and Jaboticabal season 2021–2022).

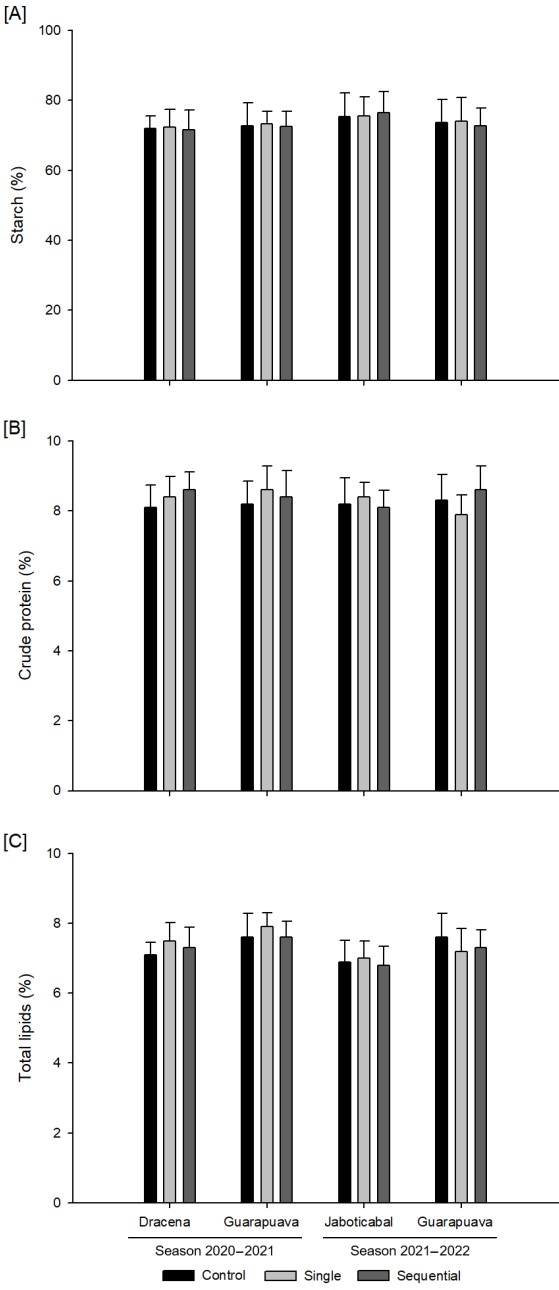

**Figure 4.** Content of starch (**A**), crude protein (**B**), and total lipids (**C**) of grain from GR maize grown with no glyphosate application (control), with a single application of glyphosate at 980 g of a.e. ha$^{-1}$ at stage V5 (single) and a sequential application of glyphosate at 520 + 980 g of a.e. ha$^{-1}$ at stage V3 and stage V7, respectively (sequential), in different locations and seasons. Error bars are one SE of the mean from ten replicates.

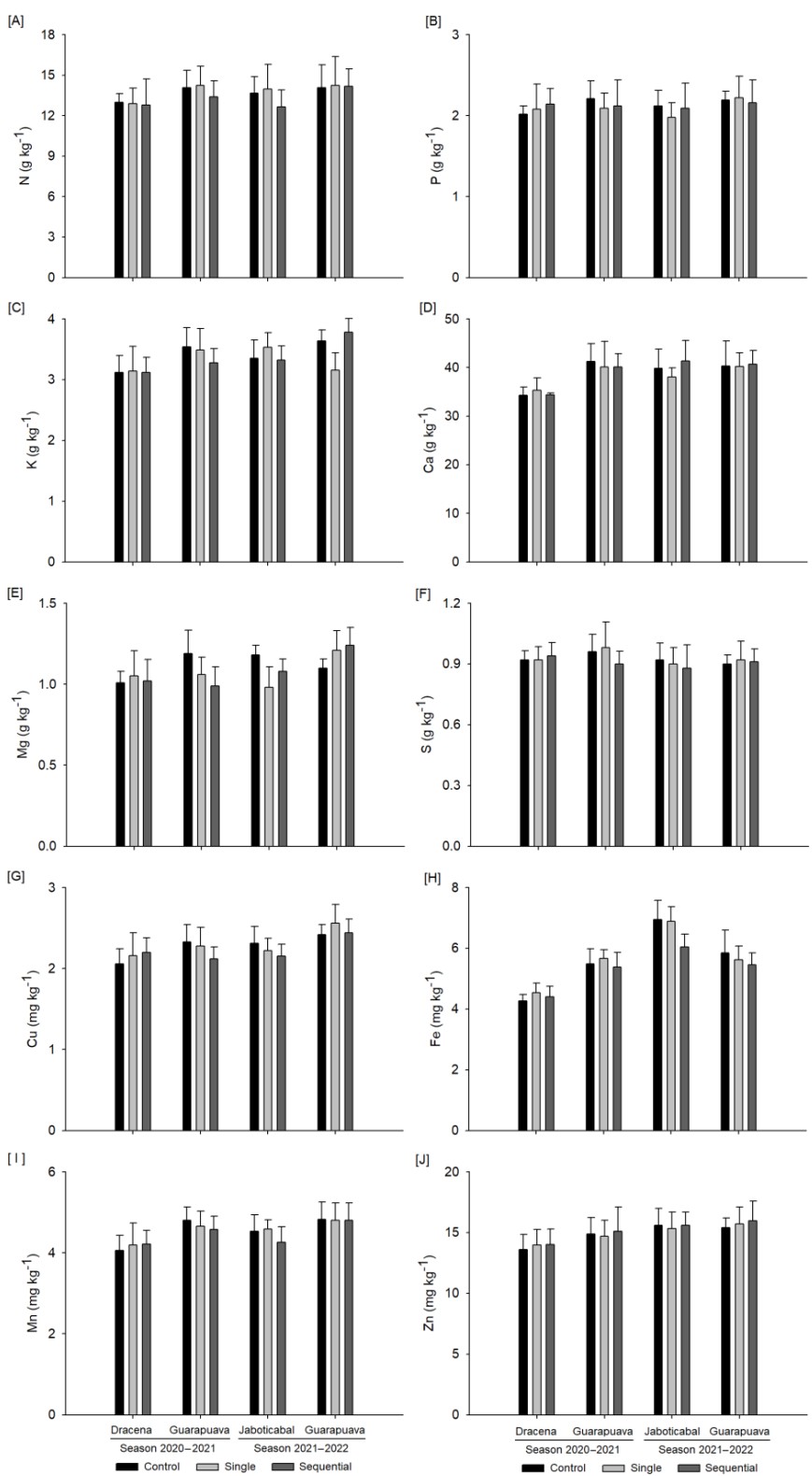

**Figure 5.** Content of macro (N, P, K, Ca, Mg, and S, respectively (**A–F**)) and micronutrients (Cu, Fe, Mn, and Zn, respectively (**G–J**)) in grain tissues of glyphosate-resistant maize growing with no herbicide application (control), with a single application of glyphosate at 980 g of a.e. ha$^{-1}$ at stage V5 (single) and a sequential application of glyphosate at 520 + 980 g of a.e. ha$^{-1}$ at stage V3 and stage V7, respectively (sequential), in different locations and seasons. Error bars are one SE of the mean from ten replicates.

### 3.5. Glyphosate and AMPA Residues in Grain

Low concentrations of glyphosate (91.3 and 101.2 ng g$^{-1}$) were detected in two of ten samples from Guarapuava from one season (2021–2022) with the sequential glyphosate application, giving a mean value of ca.20 ng g$^{-1}$ (Figure 6). No AMPA was detected in any samples.

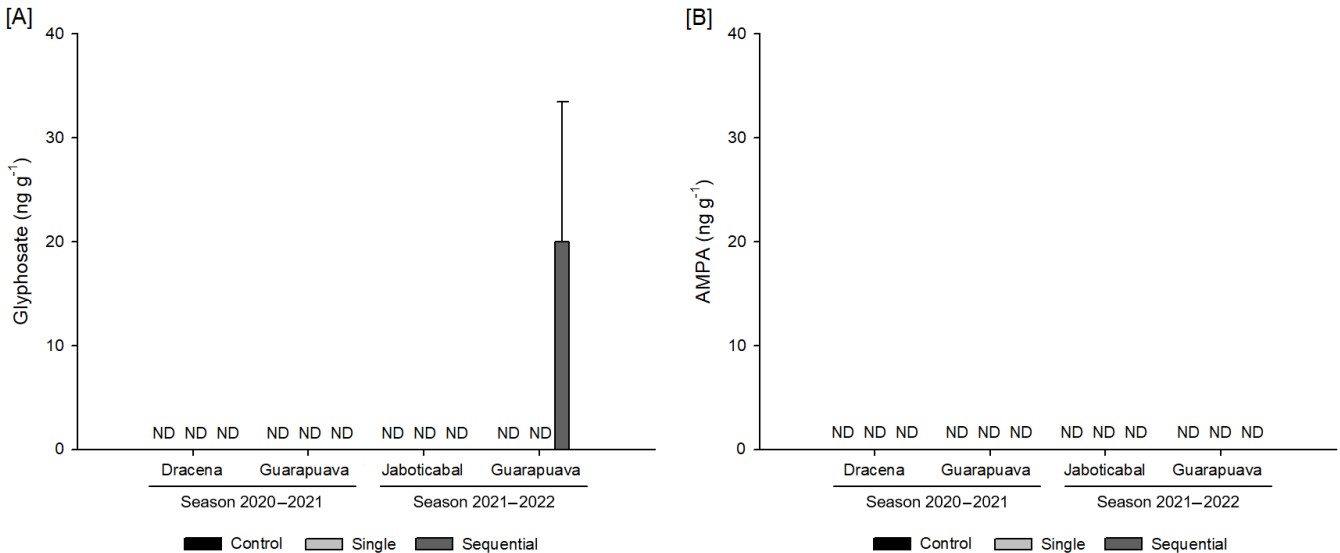

**Figure 6.** Content of glyphosate (**A**) and AMPA (**B**) in grain tissues of GR maize grown with no herbicide application (control), with a single application of glyphosate at 980 g of a.e. ha$^{-1}$ at stage V5 (single) and a sequential application of glyphosate at 520 + 980 g of a.e. ha$^{-1}$ at stage V3 and stage V7, respectively (sequential), in different locations and seasons. ND = non-detected. Error bars are one SE of the mean from ten replicates.

### 4. Discussion

Compared to GR soybeans, there are relatively few publications on the effects of glyphosate on GR maize. For example, GR soybeans have been shown to be fifty-fold more resistant to glyphosate than a near-isogenic susceptible variety [28]. In the same paper, GR canola was also found to be fifty-fold more resistant to glyphosate than a near-isogenic susceptible variety. Chinnarounder et al. [29] estimated GR maize to be 100-fold more resistant than susceptible varieties, and Hetherington et al. [30] provided data for the same resistance factor in maize. GR maize is clearly highly resistant to glyphosate. For example, Mahoney et al. [31] found foliar application of glyphosate at up to 7.2 kg ha$^{-1}$ to have no effect on yield. In a recent paper, de Araújo et al. [32] found that glyphosate applied at the V3 stage to GR maize grown in Brazil had no effect on yield or P, K, S, Fe, Cu, and Zn content up to 8.64 kg ha$^{-1}$, even though some crop injury was measured at application rates of 4.32 kg ha$^{-1}$ and higher within 21 days after application. In China, Yu et al. [33] recently reported that glyphosate applications up to 3.6 kg ha$^{-1}$ had little effect on GR maize height and no effect on ear length, ear height, nor weight of 100 grains. They only reported yield data for 0.9 kg ha$^{-1}$ treatments, for which there was no effect on yield. This work is difficult to compare with earlier work, as their transgene construct with a resistant form of 5-enolpyruvylshikimate-3-phosphate synthase (EPSPS), the target protein of glyphosate, was different from that used in the commercial varieties of other studies. Nevertheless, there was a high level of resistance. Earlier work also found no effect for 1.2 kg glyphosate ha$^{-1}$ on N, P, K, Ca, Mg, S, B, Mn, and Zn [34] nor 0.87 kg glyphosate ha$^{-1}$ on a large number of micro and macro elements [15] in GR maize. Likewise, with glyphosate applications rates similar to those of this paper, no effects were found on the GR maize mineral content nor on plant growth or yield in a greenhouse study and in a one season field study at one location [16]. At a high glyphosate application rate (2 kg ha$^{-1}$), de

Oliveria Neto et al. [35] found that GR maize root length and volume were slightly reduced, an effect that could affect mineral uptake, although shoot biomass and leaf area were slightly increased by this treatment. Another recent paper [36] reported that glyphosate had no effects on the photosynthesis of GR maize in a greenhouse study.

Hormesis (the stimulatory effect of a subtoxic level of a toxin) is common with glyphosate [37–39]. In the many reports of this phenomenon in susceptible crops and other plant species mentioned in these reviews, the application rates that stimulate growth were very low, tending to be in a narrow range just below the rate that has no effect (e.g., 2 to 30 g ha$^{-1}$). These rates are far below the usual recommended field application rates of 500 to almost 2000 g of a.e. ha$^{-1}$. A non-toxic dose of glyphosate for a GR crop can be quite high, so one might expect that hormesis could occur with some recommended doses of glyphosate on GR crops. In fact, recommended doses of glyphosate did stimulate growth in one study with GR maize [23], in which 1.7 kg ha$^{-1}$ was used and another with GR canola [40] in which 1 to 3.33 kg ha$^{-1}$ was used. Hormesis has not been reported in other studies concerning the effects of glyphosate on GR crops, nor did we find any evidence of it in this study. This indicates that the glyphosate rates that we used are probably significantly below the highest no-effect rate.

One could argue that even though glyphosate does not appear to harm GR maize at the recommended application rates in short-term experiments, there could be long-term effects of continuous growing of GR crops by indirect, adverse effects of glyphosate on soil microbes. However, no cumulative effects of glyphosate use were found on soil microbial communities of GR maize and soybean crops in a two-year study at two locations [41] and maize in a three-year study [42].

During the early days of GR crops, there were several papers from Monsanto showing that grain constituents from GR crops are substantially compositionally the same as non-GR crops, for e.g., in [26,43,44]. The crops in these studies were not treated with glyphosate, and they never published studies of the composition of grain from glyphosate-treated GR crops. Glyphosate treatment was found to have no effect on free amino acids and protein amino acids of GR soybean in a two-year, two site study [12]. In a one-year study, glyphosate had no effects on the starch, protein, carotenoid, and lipid content of GR maize [16]. The results of the current study bolster these previous findings, with no effects for the recommended glyphosate applications on the composition of GR maize seeds at three locations, one of which was for two seasons.

Contrary to the claims of Cuhra [45], due to recent claims of adverse human health effects of glyphosate, the amount of glyphosate and AMPA residues in harvested grains of GR crops has been of great interest, for e.g., in [46]. A recent review [25] summarized what is known of glyphosate and AMPA residues in the grain of GR maize treated with glyphosate. Levels of glyphosate in glyphosate-treated maize have been reported to range from undetectable levels to less than 1 µg g$^{-1}$ [16,47,48], values that are much lower than levels found in glyphosate-treated GR soybean [12,49–51] and cotton [47] seed. Only trace amounts or no AMPA were reported in the grain of GR maize treated with glyphosate by Reddy et al. [15], although very low levels can sometimes be found in the leaves of glyphosate-treated GR maize [15,52]. Rodrigues et al. [47] could not detect AMPA in two glyphosate-treated GR maize cultivars during most years and in most study sites but found low levels in some years at some sites.

The finding of no AMPA or only trace amounts in the seeds of glyphosate-treated GR maize is not surprising, as a study of AMPA accumulation in several plant species treated with glyphosate concentrations that inhibited growth by 50% at 7 days after treatment found no AMPA in either GR or non-GR maize, as well as the only other Poaceae (Gramineae) species evaluated, Italian ryegrass (*Lolium multiflorum* Lam.) [53]. AMPA was found in most of the other species, which were dicotyledonous crops and weeds. Grass weeds and maize may have very small amounts or may entirely lack the enzyme that converts glyphosate to AMPA and glyoxylate. Hearon et al. [54] reported AMPA to be readily taken up by GR maize from soil treated with either glyphosate of AMPA. The

relatively short half-life of glyphosate in soil is due to microbial degradation, mainly down to AMPA and glyoxylate [25]. Thus, all AMPA found in GR maize is not necessarily from the degradation of glyphosate in the plant, as it could be taken up from soil. However, Komoßa et al. [55] found that microbe-free maize cell cultures can metabolize glyphosate into AMPA. Nevertheless, species of Poaceae, like maize generally, have relatively little capability for degrading glyphosate [56]. It is surprising that the grass weeds (*Lolium rigidum* and *Echinochloa colona*) have evolved higher levels of degradation of glyphosate via an aldo-keto reductase as a resistance mechanism [57,58]. This enzyme has not been shown to account for the rapid degradation of glyphosate to AMPA in soybean. Under certain environmental conditions, glyphosate application to GR soybeans causes temporary chlorosis, a phenomenon called "yellow flash" by U.S farmers. Evidence points to glyphosate-caused yellow flash being caused by high AMPA accumulation [59]. AMPA is mildly phytotoxic [60], and the same symptoms are seen in foliar tissues of GR soybean containing the same amount of AMPA, whether AMPA or glyphosate-treated [59]. Both GR and non-GR crops are equally susceptible to AMPA, as it does not act by inhibition of EPSPS. The fact that no yellow flash symptoms have been reported in glyphosate-treated GR maize is almost certainly due to the lack of significant accumulation of AMPA.

All glyphosate levels found in this study were far below the allowable contamination level for maize by glyphosate (5 µg g$^{-1}$), established as an international standard by the Food and Agriculture Organization of the United Nations and the World Health Organization [61], and 1 µg g$^{-1}$ by the European Food Safety Agency, the strictest food regulatory agency in the world [62]. The concentrations that we found in two out of ten grain samples at one site in one growing season is 2% of this allowable level. The very low or no levels of glyphosate found in the seeds of GR maize is surprising because glyphosate translocates well to all metabolic sinks, such as developing seeds [10]. Phloem translocation to metabolic sinks such as meristems contributes significantly to the efficacy of glyphosate as an herbicide [63]. Phloem transport is part of the explanation for the relatively high levels of glyphosate found in the seeds of GR soybean. However, recommended application times for glyphosate in maize crops apparently occurs when translocation to metabolic sinks other than developing seeds occurs. In the present study, the latest application was performed at the V7 stage, which is before tassels are present. In previous studies, levels of glyphosate in leaf tissues of glyphosate-treated (93 g ha$^{-1}$) GR maize were much lower at 7 days after treatment than in some other species (e.g., kudzu; *Pueraria montana* var. *lobata*) given similar treatments [53], indicating that the translocation of glyphosate-treated leaves of maize may be relatively rapid. Hetherington et al. [30] found most of the glyphosate applied to GR maize to be translocated to developing shoot tissues (meristems and young leaves) and roots during a period of 7 days after treatment. Glyphosate can be lost from treated plants by root exudation [64], but this process has not been studied in GR maize. Kremer and Means [21] speculated that the changes in GR maize root rhizosphere microbes caused by foliar glyphosate treatment could be due to the exudation of glyphosate from the roots. A study of the complete movement of applied glyphosate in GR maize from application to harvest, accounting for all of the glyphosate taken up by leaves, is needed to explain the low glyphosate levels found in seeds.

## 5. Conclusions

In conclusion, our findings provide the most rigorous evidence available that recommended application rates of glyphosate do not cause adverse effects on growth, mineral composition, yield, or seed food value of GR maize under most conditions. These results are similar to the most recent previously published results with GR soybeans. Furthermore, the grain from glyphosate-treated GR maize has zero-to-trace levels of glyphosate and no AMPA, far below the maximum allowable levels set by rigorous regulatory agencies.

**Author Contributions:** Conceptualization, L.B.d.C.; methodology, V.S.B., J.C.d.S.N., C.D.d.G.M., L.T., C.A.C. and L.B.d.C.; formal analysis, S.O.D. and L.B.d.C.; resources, L.B.d.C.; data curation, L.B.d.C.; writing—original draft preparation, S.O.D. and L.B.d.C.; writing—review and editing, S.O.D. and L.B.d.C.; funding acquisition, L.B.d.C. All authors have read and agreed to the published version of the manuscript.

**Funding:** This research was funded by the National Council for Scientific and Technological Development (CNPq, Brazil, project n. 424816/2018-0) and São Paulo Research Foundation (FAPESP, Brazil, project n. 2021/04290-0). Stephen O. Duke was funded in part by the United States Department of Agriculture (USDA) Cooperative Agreement 58-6060-6-015 Grant to the University of Mississippi.

**Institutional Review Board Statement:** Not applicable.

**Informed Consent Statement:** Not applicable.

**Data Availability Statement:** The raw data supporting findings in this paper are available within the article.

**Conflicts of Interest:** The authors declare no conflict of interest.

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
