# Peer review of "Lack of Significant Effects of Glyphosate on Glyphosate-Resistant Maize in Different Field Locations"

_agronomy, doi:10.3390/agronomy13041071_

Round 1

Reviewer 1 Report

In this paper, Vitor Bidóia et al., investigated if the application of recommended rates of glyphosate (single and sequential) were used on Glyphosate- Resistant maize have significant effects on growth, foliar mineral content, crop yield,  grain quality, in the level of  residues in grain of glyphosate and aminomethylphosphonic acid (AMPA).
The research is well designed, the applied methodology is adequate, the results are presented clearly, and the discussion is adequate.
Indeed, the research answers to all five questions.

Discussion : lines 332-334.suggestion: join EFSA (European Food Safety Agency) the most strict food agency in the world

In Science, the item Conclusion is too ambitious (first paragraph - we don't know what will be published today, this week, or next week,....). I do recommend the first paragraph be rewritten.

Suggestion: Although it is not the culture object of the study, since soybean is used in the discussion, place it in the conclusions as well.

Only minor technical errors were observed

Pag.

70 – 42%-clay – 42% clay,

71 – sande – sand

100 – tem – stem

133 – 5 – g – 5 g

161 – diameterwith – diameter with

174 – kg-1-  - kg-1

189 -  (Figure 3). ). - (Figure 3).

194 – yeild – yield

214 – application (Control) – Type of letter

299 – 16,46,47  - 16, 46, 47

310 –  Italian ryegrass is Lolium multiflorum L.

References

Join DOI to the references, when they exist.

38. Jalal, A., …………………2021,207,111225. https://doi.org/10.1016/j.ecoenv.2020.111225.

Author Response

RESPONSES TO REVIEWERS IN BOLD CAPITALS

Reviewer 1

Discussion : lines 332-334.suggestion: join EFSA (European Food Safety Agency) the most strict food agency in the world ADDED

In Science, the item Conclusion is too ambitious (first paragraph - we don't know what will be published today, this week, or next week,....). I do recommend the first paragraph be rewritten. WE DISAGREE. THE EVIDENCE IN THIS PAPER AND OTHERS, PLUS THE HIGH YIELD OF GR MAIZE, IS STRONG EVIDENCE FOR A CONCLUSION. WE ADDED A FEW WORDS TO THE CONCLUSIONS.

Suggestion: Although it is not the culture object of the study, since soybean is used in the discussion, place it in the conclusions as well. WE HAVE DONE THIS

Only minor technical errors were observed

 Pag.

 70 – 42%-clay – 42% clay, CHANGE MADE

71 – sande – sand CHANGE MADE

100 – tem – stem CHANGE MADE

133 – 5 – g – 5 g CHANGE NOT MADE – AS WRITTEN, IT DIFFERENTIATES BETWEEN FIVE SAMPLES OF ONE GRAM AND SAMPLES OF 5 GRAMS

161 – diameterwith – diameter with CHANGE MADE

174 – kg-1-  - kg-1 CHANGE MADE

189 -  (Figure 3). ). - (Figure 3). CHANGE MADE

194 – yeild – yield CHANGE MADE

214 – application (Control) – Type of letter CHANGE MADE

299 – 16,46,47  - 16, 46, 47 CHANGE MADE

310 –  Italian ryegrass is Lolium multiflorum L. CHANGE MADE

 References

 Join DOI to the references, when they exist. DONE

  1. 38. Jalal, A., …………………2021,207,111225. https://doi.org/10.1016/j.ecoenv.2020.111225.

Reviewer 2 Report

Article untitled „Lack of Significant Effects of Glyphosate on Glyphosate-Resistant Maize in Different Field Locations” is a very broad and threaded manuscript, however this is another article written by the authors on a very similar topic. The manuscript presents very well and accurately results based on the effect of glyphosate on the composition of glyphosate-resistant corn samples and the quality and yield of corn crops. List of references used is impressive.   Abstract, methodology, statistical processing, presentation of the results are impeccable. The article is a laudatory sonata on glyphosate. Please explain in the Introduction and Discussion why so much research on glyphosate is being conducted. Currently, it is an indispensable element of almost each crop in the world. If the glyphosate (and AMPA) are almost not detected in plant material what happen with glyphosate?  You have written that “glyphosate translocates well to all metabolic sinks”. What is “the other side of the coin” of its use?   Why the allowable concentration of glyphosate is established by International Standard of Food Agriculture Organization of the United Nations and the World Health Organization on 5 μg g−1? The article has been presented very carefully and logically, however the re is a gap - where and how the so-called control samples were collected? Whether traditional fertilizers were used in cultivation “glyphosated crops” and control crops, what was their composition? Were the control samples taken from the same genetic type of maize. How far was the control field from the glyphosate-treated fields?

Author Response

Reviewer 2

responses are capitalized in bold

Article untitled „Lack of Significant Effects of Glyphosate on Glyphosate-Resistant Maize in Different Field Locations” is a very broad and threaded manuscript, however this is another article written by the authors on a very similar topic. The manuscript presents very well and accurately results based on the effect of glyphosate on the composition of glyphosate-resistant corn samples and the quality and yield of corn crops. List of references used is impressive.   Abstract, methodology, statistical processing, presentation of the results are impeccable. The article is a laudatory sonata on glyphosate.

Please explain in the Introduction and Discussion why so much research on glyphosate is being conducted. Currently, it is an indispensable element of almost each crop in the world. DONE

If the glyphosate (and AMPA) are almost not detected in plant material what happen with glyphosate? GOOD QUESTION. WE HAVE ADDED SOME DISSUSION ON THIS.

 You have written that “glyphosate translocates well to all metabolic sinks”. What is “the other side of the coin” of its use? WE ARE NOT SURE WHAT THE REVIEWER IS ASKING. THE FACT THAT GLYPHOSATE TRANSLOCATES WELL TO ALL METABOLIC SINKS HAS BEEN WELL ESTABLISHED FOR DECADES. WE HAVE ADDED A STATEMENT ABOUT WHY THIS CONTRIBUTES TO THE EFFICACY OF GLYPHOSATE. WE HAVE ADDED QUITE A BIT MORE ABOUT TRANLOCATION OF GLPHOSATE IN MAIZE WHERE THE GLYPHOSAE TRANSLOCATES TO.

 Why the allowable concentration of glyphosate is established by International Standard of Food Agriculture Organization of the United Nations and the World Health Organization on 5 μg g−1? WE ARE NOT REGULATORY SCIENTISTS AND DO NOT KNOW THE ANSWER TO THIS QUESTION. THIS NUMBER IS ALMOST CERTAINLY BASED ON A COMBINATION OF ACUTE AND CHRONIC TOXICITY STUDIES, AS WELL AS EXPECTED HUMAN CONSUMPTION OF MAIZE. WE ADDED THE EFSA MAXIMUM RESIDUE LEVEL TO THE PAPER, WHICH IS LOWER.

The article has been presented very carefully and logically, however there is a gap - where and how the so-called control samples were collected?  THE CONTROL PLANTS WERE GROWN IN THE SAME LOCATIONS AS THE TREATED PLANTS. THEY WERE NOT SPRAYED WITH GLYPHOSATE. THIS IS MADE CLEARER IN THE M&M NOW.

 Whether traditional fertilizers were used in cultivation “glyphosated crops” and control crops, what was their composition? WE USED DIFFERENT FERTILIZERS IN EACH LOCATION AND YEAR, AIMING TO PROVIDE A HIGH-PRODUCTIVITY. FOR EXAMPLE, IN JABOTICABAL, WE USED 8-32-16 (NPK) THE FIRST YEAR AND 4-14-8 (NPK) THE SECOND YEAR. SINCE THIS IS IRRELEVANT TO THE STUDY, WE HAVE LEFT THIS INFORMATON OUT.

 Were the control samples taken from the same genetic type of maize. YES, THE CONTROLS WERE THE EXACT SAME GR VARIETY AS THE TREATED. THE OBJECTIVE WAS NOT TO COMPARE VARIETIES BUT TO DETERMINE THE EFFECTS OF GLYPHSOATE ON A SINGLE VARIETY.

How far was the control field from the glyphosate-treated fields? THE CONTROL TREATMENTS WERE RANDOMIZED WITH THE GLYPHOSATE TREATMENTS WITHIN BLOCKS, IN EACH LOCATION. FOR GLYPHOSATE APPLICATION, WE PROTECTED THE SIDES OF THE PLOTS USING PLASTIC TARPS TO PREVENT HERBICIDE DRIFT TO OTHER PLOTS. THE CONTROL SAMPLES WERE COLLECTED FROM THE TWO CENTRAL LINES OF THE CONTROL PLOTS. THIS INFORMATION WAS ADDED TO THE TEXT.

Reviewer 3 Report

Well prepared paper. It looks for the future problem and discussion about glyphosate use. 

My suggestions are:

Line 100: plant stem

Line 118: obtained

Line 124: the weight of one thousand grains (not hundred) because the weight is more than 300 grams

Line 134: method, (no dot)

Line 161: diameter with

Line 187: thousand grains

Line 191: thousand  not hundred grams but grains

Figure 3B: Weight of one thousand grains

Line 197: thousand grains (B)

Line 214: application (Control) - different fonts

Line 250: of 1000 grains.

Line 368: use soybeans (without  dot) 

Line 395: Filho, R.V., (without comma)

Line 417: Brassica

Line 434: change = to -

Line 472: Environ. Sci.

Author Response

Reviewer 3

responses are capitalized in bold

Well prepared paper. It looks for the future problem and discussion about glyphosate use. 

My suggestions are:

Line 100: plant stem  CHANGE MADE

Line 118: obtained  CHANGE MADE

Line 124: the weight of one thousand grains (not hundred) because the weight is more than 300 grams WE ARE GLAD YOU FOUND THIS ERROR – CHANGE MADE

Line 134: method, (no dot) CHANGE MADE

Line 161: diameter with CHANGE MADE

Line 187: thousand grains CHANGE MADE

Line 191: thousand  not hundred grams but grains CHANGE MADE

Figure 3B: Weight of one thousand grains  CHANGE MADE

Line 197: thousand grains (B)  CHANGE MADE

Line 214: application (Control) - different fonts CHANGE MADE

Line 250: of 1000 grains. CHANGE MADE

Line 368: use soybeans (without  dot)  CHANGE MADE

Line 395: Filho, R.V., (without comma) CHANGE MADE

Line 417: Brassica CHANGE MADE

Line 434: change = to - CHANGE MADE

Line 472: Environ. Sci. CHANGE MADE